# Textile Display with AMOLED Using a Stacked-Pixel Structure on a Polyethylene Terephthalate Fabric Substrate

**DOI:** 10.3390/ma12122000

**Published:** 2019-06-22

**Authors:** Jae Seon Kim, Chung Kun Song

**Affiliations:** 1National Disaster Management Research Institute, Ulsan 44548, Korea; js9996@nate.com; 2Department of Electronics Engineering, Dong-A University, Busan 49315, Korea

**Keywords:** electronic textiles, AMOLED, OTFTs, OLEDs, textile displays, organic thin film

## Abstract

An active-mode organic light-emitting diode (AMOLED) display on a fabric substrate is expected to be a prominent textile display for e-textile applications. However, the large surface roughness of the fabric substrate limits the aperture ratio—the area ratio of the organic light-emitting diode (OLED) to the total pixel area. In this study, the aperture ratio of the AMOLED panel fabricated on the polyethylene terephthalate fabric substrate was enhanced by applying a stacked-pixel structure, in which the OLED was deposited above the organic thin-film transistor (OTFT) pixel circuit layer. The stacked pixels were achieved using the following three key technologies. First, the planarization process of the fabric substrate was performed by sequentially depositing a polyurethane and photo-acryl layer, improving the surface roughness from 10 μm to 0.3 μm. Second, a protection layer consisting of three polymer layers, a water-soluble poly-vinyl alcohol, dichromated-polyvinylalcohol (PVA), and photo acryl, formed by a spin-coating processes was inserted between the OTFT circuit and the OLED layer. Third, a high mobility of 0.98 cm^2^/V∙s was achieved at the panel scale by using hybrid carbon nano-tube (CNT)/Au (5 nm) electrodes for the S/D contacts and the photo-acryl (PA) for the gate dielectric, enabling the supply of a sufficiently large current (40 μA @ V_GS_ = −10 V) to the OLED. The aperture ratio of the AMOLED panel using the stacked-pixel structure was improved to 48%, which was about two times larger than the 19% of the side-by-side pixel, placing the OLED just beside the OTFTs on the same plane.

## 1. Introduction

Electronic textiles (e-textiles) are attracting much attention because they are expected to provide new functionality by integrating electronic devices into textiles [1,2]. Especially, as wearable electronics and the Internet of Things are emerging as prominent killer applications of IT technology [3], e-textiles are becoming an increasingly important technology because of their greater convenience and functionality than conventional hand-carried devices. It is expected that e-textiles will be applied in very diverse areas such as healthcare, sports, fashion, and the military [1,2].

Technologies related to e-textiles have evolved from a gadget style, where electronic circuit boards were simply attached onto textile surfaces [4,5,6,7,8], to on-cloth applications, where electronic devices are directly fabricated onto the textiles [1]. In the future, in-cloth textiles will be developed, where active devices such as organic thin-film transistors (OTFTs), organic light-emitting diodes (OLEDs), and organic photovoltaics will be implemented on a single fiber and the active fibers will be woven together to realize electronic functions in textiles [9,10,11,12,13,14].

A display is a key device needed in order to realize the ubiquitous features of e-textiles; information should be able to be obtained at any place and any time. Among the currently available displays, an OLED is a unique device which can be implemented on textiles because of the attractive properties such as self-emission, the capability of low-cost solution processes, and the possible applications to flexible electronics, enabling to use a variety of substrates [15,16,17,18,19]. However, although articles about OLEDs on fabric substrates [20] and on a single fiber [21] have been published, reports regarding electronic display panels integrating OLEDs and OTFTs on a fabric substrate are rare. 

Previously, the authors of this paper published an article regarding an active-mode organic light-emitting diode (AMOLED) panel fabricated on a fabric substrate [22]. In the paper, the AMOLED panel adopted a side-by-side pixel structure having an OLED beside the OTFTs’ pixel circuit on the same plane. However, the aperture ratio—the area ratio of the OLED to the whole pixel—could not be larger than 20% due to the large area of the OTFTs required to drive the OLED and the rough surface of the fabric substrate. 

In this paper, to improve the aperture ratio of the AMOLED display, a stacked-pixel structure placing the OLED above the OTFTs’ circuit layer was developed for the fabric substrate. With the stacked-pixel structure, the aperture ratio was increased by approximately 2.5 times that of the side-by-side pixel structure.

## 2. Design of the AMOLED Panel with a Stacked-Pixel Circuit

In this study, the AMOLED panel adopted a standard pixel circuit, consisting of two OTFTs, one OLED, and one capacitor, as shown in Figure 1a. The switching OTFT (SW OTFT) was activated by applying a scan voltage (V_scan_) to the gate of the SW OTFT. Then, information on the data line (V_data_) was transferred to the storage capacitor (C_st_) through the SW OTFT. The key function of the SW OTFT was to supply a sufficiently large current in the on-state to the C_st_, and thus, to cause the voltage across the C_st_ to increase quickly during the scanning period. Having an extremely low off-state current in the off-state was also an important feature to sustain the voltage on the C_st_ during the time frame. The driving OTFT (DR OTFT) was activated by the voltage stored in the C_st_. The DR OTFT should supply a large current to the OLED to light it up brightly.

Since the mobility of the OTFTs was less than 1 cm^2^/V∙s, a large ratio of the channel-width-to-length (W/L) was required to supply a large current, as described above. Therefore, the aperture ratio was generally less than 20% when the OTFTs and OLED were placed on the same plane; this is called a side-by-side structure [22].

In this study, the aperture ratio was enhanced by employing a stacked-pixel structure, where the OLED was placed above the OTFTs’ pixel circuit, as shown in Figure 1b. In order to realize the stacked structure in the panel, several technological issues needed to be resolved. First, to supply a sufficiently large on-state current (I_on_) to the enlarged OLED in the stacked pixel, the OTFTs’ performance needed to be improved, because the OLED area relative to the OTFTs’ circuit was larger compared to a side-by-side structure. Next, a protection layer (PL) should be inserted between the OTFTs’ circuit layer and the OLED layer in order to protect the OTFTs’ circuit from being damaged by the OLED processes performed above it. The PL must not affect the OTFTs below and should also have a self-patterning ability. Otherwise, a patterning process, such as photo-lithography, may seriously damage the OTFTs. In the next section, the fabrication processes will be described by focusing on these issues. 

Based on a channel length of L = 20 μm as a minimum feature size, the AMOLED panel was designed as having 64 × 64 pixels. The channel width of the DR OTFT and the SW OTFT were designed to be W = 180 × L and 34 × L, respectively. Therefore, the pixel pitch was 1 mm × 0.77 mm, with an aperture ratio of 48%, where the OLED area was 0.76 mm × 0.49 mm, and a panel diagonal length of 3.2 inches. In the stacked pixel, the pixel pitch was reduced by about 70%, meanwhile, the aperture ratio was increased by 2.5 times compared with a side-by-side pixel with the same minimum feature size.

## 3. Fabrication

The fabrication processes are depicted in Figure 2a. Polyethylene terephthalate (PET) fabric was used as a substrate. The PET fabric was woven with PET fibers with a diameter of 200 μm. Before starting the processes, the PET fabric was pre-shrunken by heating at 150 °C for 3 h to avoid deformation as well as to minimize misalignment due to the shrinkage caused by the subsequent thermal processes. Then, the PET fabric was attached to a carrier glass using UV-detachable glue, and the edge sides of the substrate were strongly fixed on the glass with adhesive tape. This prevented the various chemicals from swelling the fabric during the subsequent processes and also the substrate from being deformed under high-temperature processes.

The rough surface of the PET fabric was smoothed by a special planarization process using a double layer of polyurethane (PU) and photo-acryl (PA) [22,23]. The surface roughness was reduced from 10 μm to 0.3 μm, which was suitable for fabricating devices on, because the high surface roughness was likely to disconnect thin devices, meanwhile, a small surface roughness can be continuously covered with thin OTFT and OLED devices. Subsequently, aluminum was evaporated on the smoothed fabric and patterned by photo-lithography for the gate electrodes of the OTFTs and scan bus lines. In addition, PA was spin coated for the gate dielectric layer of the OTFTs. The performance of the OTFTs strongly depended on the compositional ratio of the solvent in PA solution, and thus, a proper ratio was determined in order to produce a high performance. The results are discussed in the next section. Since the PA had a self-patterning ability, the layer was patterned for the gate area by exposing it to UV through a mask without a photoresist process. 

As described in the previous section, the OTFT performance needed to be improved in order to supply a large on-current to the enlarged OLED in the stacked AMOLED. Therefore, the source and drain (S/D) contacts of the OTFTs used hybrid electrodes consisting of carbon nanotubes (CNTs) and Au because they produced the lowest contact resistance (2.9 K∙cm) due to the work function modulation of CNTs with the deposition of Au on the CNTs. This resulted in a large on-state current. The detailed structure and the characteristics of the hybrid electrodes can be seen in Reference [24]. A CNT solution was spray-coated on the whole PA layer and then Au was evaporated with a thickness of 5 nm. The S/D electrodes were patterned using a photo-lithography process. By evaporating pentacene through a shadow mask for SW and DR OTFTs, the processes for the OTFTs’ pixel circuit layer was completed. 

As described in the previous section, a PL was deposited on the OTFTs’ circuit layer to stack the OLED above it. To achieve the requirements as described in the previous section, the PL consisted of three polymer layers, including a water-soluble poly-vinyl alcohol (w-PVA), dichromated-PVA (d-PVA), and a PA. The w-PVA was applied to protect the pentacene OTFTs from being damaged by the organic solvents from the PA. The water solution did not affect the pentacene due to their different hydrophobicity. The d-PVA was used to pattern the w-PVA. The PA protected the double PVA layers and the OTFTs from the effects of the OLED process.

The w-PVA solution was prepared by mixing well 3 wt % PVA molecules with deionized (DI) water. The PVA solution was spin-coated on the panel containing the OTFTs’ circuits at 1000 rpm for 20 s and dried for 30 min in air. Subsequently, the d-PVA solution, which was formulated by mixing ammonium dichromate of 0.03 wt % with the w-PVA solution, was spin-coated on the w-PVA film at 1000 rpm for 20 s and dried for 30 min in air. The total thickness of the PVA double layer was approximately 5 μm. The double PVA layers were exposed to UV for 1 min through a chrome mask and developed using DI water. The developed PVA film was baked at 60 °C for 10 min. Additionally, the PA solution with the same mixing ratio as the gate dielectric was spin-coated on the developed PVA film at 1000 rpm for 20 s and softly baked at 90 °C for 10 min. The thickness of the PA was about 1 μm. The PA film was patterned via-holes to interconnect the OTFTs to the OLED by exposure to UV for 30 s, developed for 40 s, and then hard-baked at 130 °C for 60 min. 

Silver was evaporated on the patterned PA layer for the anode electrodes of the OLED, and another PA was spin-coated on the Ag electrodes and patterned to define the OLED area. Subsequently, the OLED layers were sequentially evaporated through a shadow mask, and the transparent cathode electrodes were evaporated above the OLED with 4,4’-bis(N-phenyl-1-naphthylamino) NPB) (40 nm)/Ag (20 nm)/Al (1 nm) layers. Finally, an encapsulation layer consisting of w-PVA and PA was spin-coated on the panel. The final AMOLED panel was detached by exposing it to UV through the carrier glass. Figure 2b shows the detached AMOLED panel picture, including the stacked AMOLED and the side-by-side AMOLED panel for comparison and the various test elements such as the discrete OTFTs, the OLED, and pixels.

## 4. Results and Discussion

The surface roughness of the fabric substrate was measured by AFM (Park System, Seoul, Korea) at each step of the planarization process. The surface roughness was plotted and also compared with the conventional plastic and glass substrate in Figure 3. The surface roughness of the PET substrate was reduced from 10 μm to 0.3 μm (as shown in Figure 3) after the planarization process, as described in the previous section. The PU layer decreased the roughness at the macro scale from 10 μm to 2 μm, and the PA decreased the roughness at the micro scale from 2 μm of the PET/PU to 0.3 m and also enhanced the process compatibility to the subsequent films by changing the hydrophobicity of the PET/PU as well. Although the surface roughness of the PET/PU/PA substrate (0.3 μm) was larger than that of poly carbonate (PC) plastic (0.025 μm) and glass (0.003 μm) substrates, it was comparable to the thickness of OTFTs (0.43 μm) and OLEDs (0.3 μm). Therefore, the devices could be fabricated on the planarized PET/PU/PA substrate.

The hybrid electrode of CNT/Au for the S/D contacts together with PA for the gate dielectric in the OTFTs was employed for the first time for an AMOLED panel to improve performance as well as to reduce the process steps in this work. To reduce the interface states density and to obtain a smoother gate surface, the PA solution was diluted from the as-purchased state by adding the solvent of propylene glycol monomethyl ether acetate (PGMEA). The electrical characteristics were varied with the mixing ratio of PA to PGMEA, as shown in Figure 4. Representative transfer curves are depicted according to the various mixing ratios of the PA solutions. The electrical parameters are summarized in Table 1, where the values were averaged out of sixteen OTFTs for each ratio.

The OTFTs with a ratio of 1:3 produced the largest mobility of 0.98 cm^2^/V∙s, which was 49 times larger than the 0.02 cm^2^/V∙s of the as-purchased PA, as shown in Table 1, and comparable to the 1.0 cm^2^/V∙s of the OTFTs using the polyvinylphenol (PVP) gate dielectric and the CNT/Au (5 nm) for the S/D contacts, as reported in Reference [24]. The required performance enhancement for the OTFTs, as well as the reduction in process steps in the stacked AMOLED panel, were successfully achieved using PA for the gate dielectric and the CNT/Au (5 nm) electrodes for the S/D contacts. 

The high performance of the OTFTs was degraded after the PL was deposited on them. In Figure 5, the transfer curves of the DR and SW OTFTs without the PL are compared to those with the PL; the transfer curves were measured from separated test pixels. As shown in Table 2, the mobility was reduced by about 40%, from 0.87 cm^2^/V∙s and 0.75 cm^2^/V∙s to 0.54 cm^2^/V∙s and 0.49 cm^2^/V∙s for DR and SW OTFT, respectively. Although the water in the w-PVA of the PL was expected to protect the hydrophobic pentacene of the OTFTs from the subsequent processes, the developing process of the d-PVA layer and the depositing process of the PA layer seemed to damage the pentacene. However, the on-state currents at 25 V were sufficiently large with 228 μA and 7.97 μA for the DR and SW OTFT, respectively. As a result, the DR OTFT could drive the large OLED and the SW OTFT could quickly charge up the storage capacitor. In addition, the off-state current of the SW OTFT, 1.69 pA/μm, was small enough to keep the charge on the storage capacitor during the time frame. Even though the performance of the DR and SW OTFTs deteriorated after depositing the PL, the performance was still able to operate the AMOLED panel.

In Figure 6a, the structure of the phosphorescent OLED used in this paper is presented. It consists of multiple organic layers of 1,4,5,8,9,11-hexaazatriphenylene-hexacarbonitrile (HAT-CN) (10 nm)/NPB (30 nm)/4,4’-Cyclohexylidenebis[N,N-bis(4-methylphenyl)benzenamine] (TAPC)(10 nm)/4,4’-Bis(N-carbazolyl)-1,1’-biphenyl (CBP):Ir (ppy) (20 nm)/2,2’,’’-(1,3,5-Benzinetriyl)-tris(1- phenyl-1-H-benzimidazole) (TPBi)(40 nm)/LiF (0.2 nm) between the cathode and anode electrodes. It also used a reflective anode of Ag (80 nm) and a transparent cathode of Al (1 nm)/Ag (20 nm)/NPB (40 nm) to implement the top emission on the opaque fabric substrate. The various types of OLED lights were also fabricated on the planarized PET/PU fabric substrate with an area of 20 cm × 20 cm, and a representative OLED light with an area of 7 cm × 7 cm is shown in Figure 6b. They successfully operated without an electrical short, reflecting that the surface roughness of the PET/PU substrate (0.3 μm) was smooth enough for OLEDs, even when the area was large. The OLEDs produced a phosphorescent green light with a wavelength of 534 nm and a luminance of 23,673 cd/m^2^ at 7 V, as shown in Figure 6c.

In order to protect the AMOLED panel from damage due to the air exposure during measurements, a temporary encapsulation consisting of PVA and PA double layers was spin-coated with a thickness of 1 μm for each layer on the final AMOLED panel. It was expected that the water in the PVA solution would not affect the hydrophobic organic layers of the OLEDs, and the thick polymer layers would protect the OLEDs from air exposure. As shown in Figure 6d, as a layer was added, the luminance decreased from the initial luminance of 19,895 cd/m^2^ at 7 V with a PVA single layer encapsulation to 16,636 cd/m^2^ with the PVA/PA double-layer encapsulation. However, the encapsulation retarded the degradation by keeping air from permeating into the OLEDs. The luminance of the OLEDs gradually decreased to 64% of the initial luminance with the PVA/PA encapsulation and to 58% with the PVA encapsulation after 10 days in air. Meanwhile, the bare OLED decreased to 54%. The AMOLED panel with the encapsulation maintained visible brightness for 10 days, although the brightness became dim with time, as shown in the inset of Figure 6d. 

In Figure 7, two types of AMOLED panels are compared. The side-by-side pixels produced an aperture ratio of 19%, and the space between the OLEDs (1130 μm), where the DR and SW OTFTs were contained, was discernable even with bare eyes.

Meanwhile, in the stacked pixels, the pixel space could not be distinguished because the light coming out of a pixel overlapped those of the neighboring pixels. The overlapping light occurred due to the smaller pixel space (280 μm) and the brighter luminance of the OLEDs caused by the larger on-state current (40 μA @ V_GS_ = −10 V) of the DR OTFT. The aperture ratio was 48%, approximately 2.5 times larger than that of the side-by-side pixels.

The high-aperture ratio in the AMOLED panel was successfully achieved using the stacked pixel structure, which was possible due to the PL between the OTFTs’ pixel circuit and the OLED, as well as the improved performance of the OTFTs by using CNT/Au (5 nm) hybrid electrodes for the S/D contacts and the self-patterning PA gate dielectric. 

The washing problem is a large obstacle to overcome for practical applications of electronic textiles including AMOLED textile displays. Recently, articles concerning water resistant encapsulation layers for e-textiles have been reported [25,26]. They have successfully protected the underlaid OLEDs on a fabric without performance deterioration, even after being washed 10 times. Therefore, the washing issue can be resolved in the near future. 

## 5. Conclusions

In this paper, a textile display of AMOLED was successfully fabricated on a PET fabric substrate. The aperture ratio was significantly enhanced using a stacked-pixel structure having an OLED on the OTFTs’ pixel circuit. Three key technologies were employed: a planarization process on the rough PET substrate; the insertion of a protection layer (PL) between the OLED and the OTFTs’ pixel circuit; and the use of hybrid CNT/Au (5 nm) electrodes for the S/D contacts together with PA for the gate dielectric of the OTFTs. The planarization process consisting of polyurethane and photo-acryl layer reduced the roughness from 10 μm to 0.3 μm, which was smooth enough for the devices. The PL consisted of three polymer layers: a water-soluble poly-vinyl alcohol (w-PVA), dichromated-PVA (d-PVA), and photo acryl (PA). This protected the bottom OTFTs from damage by the subsequent OLED process, and also enabled patterning for the interconnection between the bottom OTFTs and the top OLED without an additional lithography process. The hybrid CNT/Au (5 nm) electrodes used for the S/D contacts together with the PA for the gate dielectric noticeably increased the on-state current of the OTFTs, which could then provide a sufficiently large current to the enlarged OLED for high luminance. With those technologies, an aperture ratio of 48% was successfully achieved, which was 2.5 times larger than the 19% of the side-by-side pixel structure. In addition, it was also possible to eliminate several photo-lithography processes in the fabrication of the AMOLED panel, which might otherwise cause damage to the organic layers, by using PA for various layers, such as the gate dielectric of the OTFTs and for the PL and the encapsulation layer.

A sample of the AMOLED textile display panel was successfully demonstrated in this paper. I In the future, textile display technology will be advanced for low power consumption, and thus, will be able to integrate with flexible batteries. Furthermore, it is expected that the washing problem will be resolved in the near future. Therefore, the future use of practical AMOLED textile displays for commercial production is not hard to see. 

## Figures and Tables

**Figure 1 materials-12-02000-f001:**
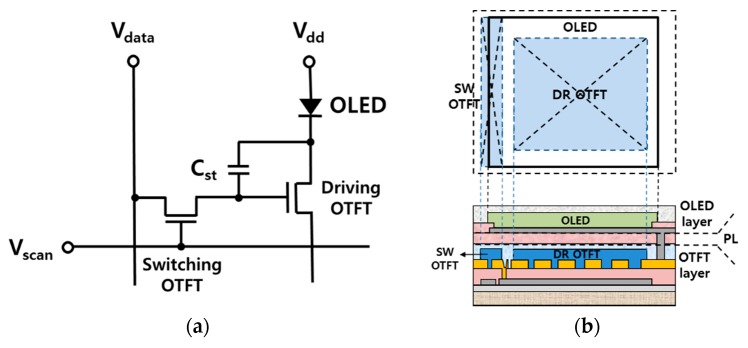
(**a**) The pixel circuit of the active-mode organic light-emitting diode (AMOLED) panel consisting of 2 organic thin-film transistors (OTFTs), 1 organic light-emitting diode (OLED), and 1 capacitor; and (**b**) the layout and cross-section of the stacked pixel.

**Figure 2 materials-12-02000-f002:**
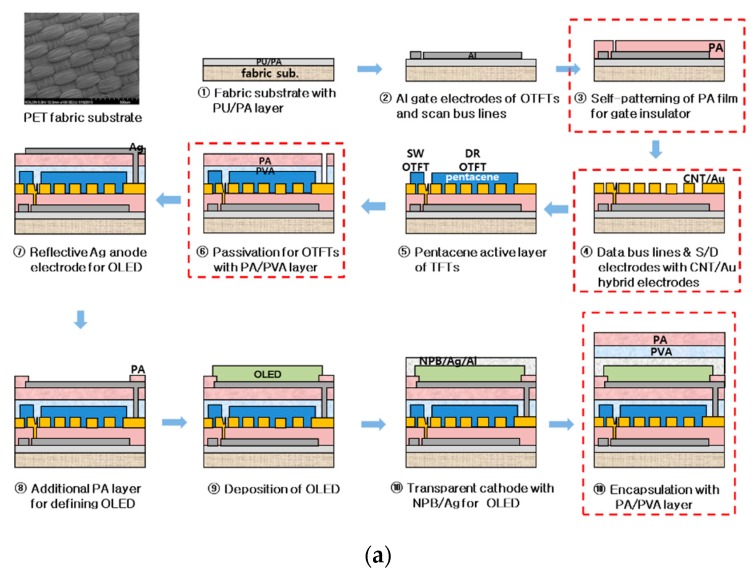
(**a**) The fabrication processes for the AMOLED panel using the stacked pixel and (**b**) a picture of two AMOLED panels, using the stacked and the side-by-side pixels, fabricated on a polyethylene terephthalate (PET) fabric substrate with the various test elements included.

**Figure 3 materials-12-02000-f003:**
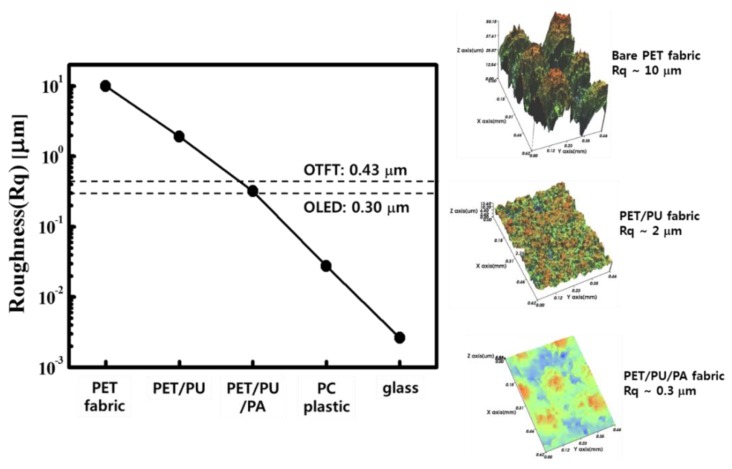
The surface roughness variation of the polyethylene terephthalate fabric substrate according to deposition of the polyurethane (PU) and photo-acryl (PA) layers, including the AFM images of bare PET, PET/PU, and PET/PU/PA fabric substrates.

**Figure 4 materials-12-02000-f004:**
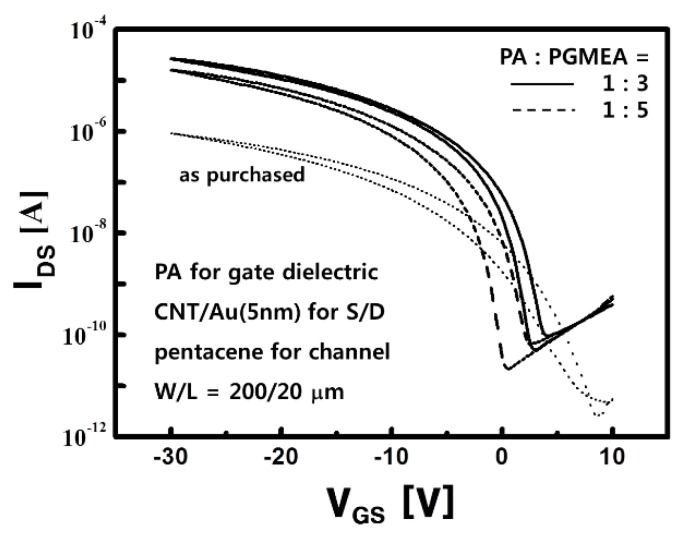
The transfer curves of the pentacene-OTFTs using photo-acryl (PA) for the gate dielectric with the various mixing ratios of PA to PGMEA, in which the CNT/Au (5 nm) electrodes were used for the S/D contacts.

**Figure 5 materials-12-02000-f005:**
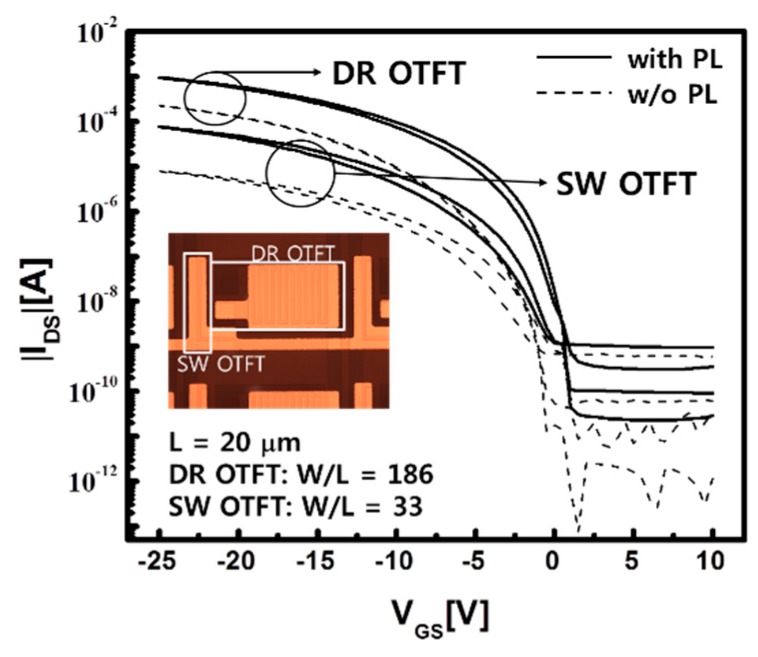
The transfer curves of the DR and SW OTFTs with and without the protection layer in the test pixel; the performance was degraded by the protection layer; however, the on-state current was still large enough to drive the enlarged OLED of the stacked pixel.

**Figure 6 materials-12-02000-f006:**
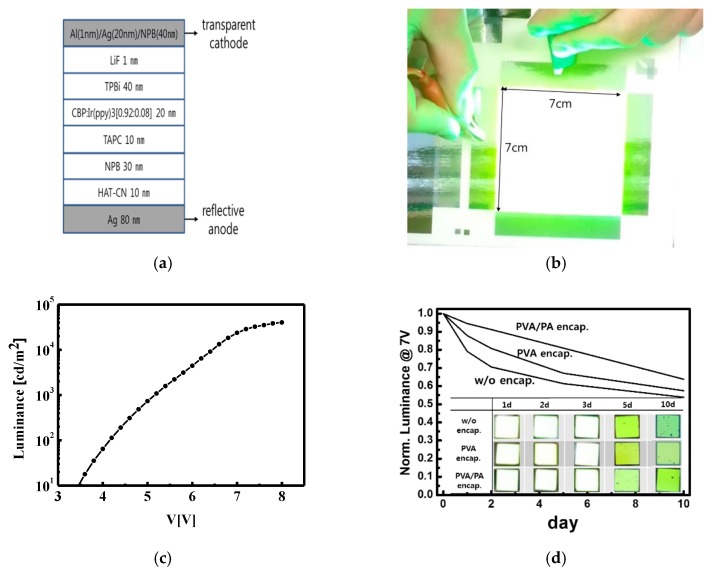
(**a**) The structure of the phosphorescent OLED, (**b**) an OLED light fabricated on a PET fabric substrate with an area of 7 cm × 7 cm, (**c**) the luminance characteristics of the top emitting phosphorescent OLED fabricated on the PET fabric substrate, and (**d**) the degradation of the OLED luminance with and without the PVA/PA encapsulation.

**Figure 7 materials-12-02000-f007:**
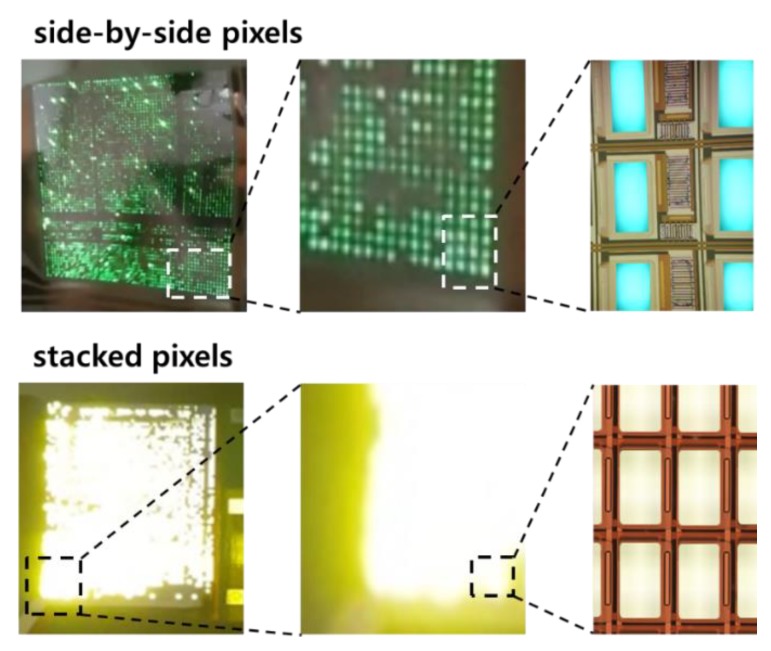
Comparison of the stacked pixels with the side-by-side pixels in the AMOLED fabricated on the PET fabric substrate; the aperture ratio of 48% with the stacked pixels (pixel pitch: 1.0 mm × 0.77 mm, OLED area: 0.76 mm × 0.49 mm) is clearly identified with 19% of the side-by-side case (pixel pitch: 1.6 mm × 1.6 mm, OLED area: 1.04 mm × 0.47 mm).

**Table 1 materials-12-02000-t001:** The average values of the electrical parameters of the pentacene OTFT using photo-acryl (PA) as the gate dielectric and the CNT/Au (5 nm) electrodes for the S/D contacts according to various mixing ratios of PA to the solvent PGMEA.

Mixing Ratio	μ_FET_ (cm^2^/V·s)	I_on_ @ −30 V (μA)	I_off_ (pA/μm)	I_on/off_	V_on_ (V)	SS (V/dec)	ε
1:1	0.02	0.91	0.02	1.9 × 10^5^	8.6	2.70	2.70
1:3	0.98	26.5	0.26	5.1 × 10^5^	4.2	1.65	2.57
1:5	0.72	15.9	0.11	7.6 × 10^5^	2.6	1.30	2.36

μ_FET_: field effect mobility; I_on_: the on-state current; I_off_: the off-state current: SS: sub-threshold slop: ε: dielectric constant.

**Table 2 materials-12-02000-t002:** The average values of the electrical parameters of the DR and SW OTFTs in the stacked pixel with and without the protection layer.

Device (W/L)	Protection Layer	μ_FET_ (cm^2^/V·s)	I_on_ @ −25 V (μA)	I_off_ (pA/μm)	I_on/off_	V_on_ (V)	SS (V/dec)
DR OTFT (186)	without	0.87	961	0.15	3.39 × 10^7^	2.0	1.21
	with	0.54	228	0.09	1.31 × 10^7^	0.0	2.30
SW OTFT (33)	without	0.75	76.8	3.27	7.11 × 10^5^	1.0	2.76
	with	0.49	7.97	1.69	1.43 × 10^5^	1.5	5.91

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
