# Peer review of "Textile Display with AMOLED Using a Stacked-Pixel Structure on a Polyethylene Terephthalate Fabric Substrate"

_materials, 2019, doi:10.3390/ma12122000_

Round 1
Reviewer 1 Report
The third and fourth paragraphs need to be rewritten with clearly without mix with the oldest articles of the authors and taking account the following points:
1. The authors need to write about the materials and instrument which had used before the third paragraph (fabrication).
2. The description of fabric specifications are messing (weaving, knitting ,...) yarn structure ( continues filament,…)
3. Line 97, what is the special planarization process (coating, printing, ….)?
4. In line 98, the double layer is from two sides of fabric or for one side, if it is from one side why we apply to layers, what is the thickness for each layer?
5. Line 99, who the measurement of surface roughness was done?
6. Line 99, the authors wrote (which was suitable for fabricating devices on it), they need to explain why 10 µm is not suitable for fabricating devices and 3 µm is suitable.
7. In line 110, the authors wrote “The detailed structure and the characteristics can be seen in the article of [11]”, That is mean the fabrication method is a part of the literature review?
8. I didn’t found in the text how the authors obtained figure 3?
9. Line 156, who the measurement has been done?
10. Who the authors obtain figure 4?
11. Table 1 is missing
12. Table 2. Is missing
Author Response
We cannot write the answers to your comments in this web site.
Please refer to the uploaded file.
Thank you!

Reviewer 2 Report
The authors developed a technology of stacked pixel structure placing OLED above the OTFTs circuit layer for fabric substrate to improve the aperture ratio of AMOLED display. The aperture ratio is increased by about 2.5 times larger than that of the side-by-side pixel. The work is interesting and the manuscript is well organized. It can be accepted after addressing the following questions:
1. The morphology of the fabric substrate should be illustrate by optical image or SEM that the readers can understand the improvement effect of the surface roughness.
2. The inset images in the figure 6 are not clear and they can be enlarged in the revision.
3. The references cited are all before 2016, more recently published work should be discussed that the readers can understand the progress better.
4. The formation of the reference should be carefully checked.
Author Response
We cannot properly writ the answers to your comments in this web site.
Please refer to the attached file below.
Thank you!

Reviewer 3 Report
Dear authors,
Your paper deals with OLED on textile. This paper is quite interesting but I would suggest the following improvements :
1°) You should develop "the state of the art" of OLED on textile. You mostly mentionned your previous works (OLED & OTFT side by side) but I would appreciate to read the achievements of the other labs. By the way, your references could be updated.
2°) Your fabrication process looks quite complex. You should mention if this process is compatible with the fabric industry.
3°) You mentionned about 10V to power supply your AMOLED's. What will be the power consumption? How do you plan to integrate the proper battery on/in the textile ?
4°) I think you should also do and discuss "preliminary reliability tests" vs the fabric uses (mechanical, washing, ...)
5°) Table 1 & Table 2 are missing
6°) Line 109 : you wrote "contact resistance" but the given value (2.9 kohms.cm) is related to a resistivity. This is not the same and you have to explain in more details why CNT's + Au improve the contact resistance. By the way, I look without success for ref [11].
7°) On Figure 4,there are 2 curves for each PA gate dielectric, Why ?
8°) Lines 174-186 : the discussion on the PL Protective Layer is a little bit confusing.
9°) In the conclusion, you should draw some perspectives
Regards,
Author Response
We cannot properly write the answers to your comments in this web site.
Please refer to the attached file.
Thank you!

Round 2
Reviewer 1 Report
No further comments.
Author Response
Thank you!
Reviewer 3 Report
Dear authors,
You partially answer to my questions and I still believe that : process compatibility with fabrics, power consumption and reliability issues are important topics that have to be taken into account.
Nevertheless I still have two comments :
- fabrication process : I don't think that photolithography technique is compatible with R2R.
- figure 4 : There are 2 curves for each conditions (as purchased, 1:3 and 1:5). Could you precise the reason ?
Regards,
Author Response
Please refer to the attached reply.
Thank you!
